# Controlled Delivery of an Anti-Inflammatory Toxin to Macrophages by Mutagenesis and Nanoparticle Modification

**DOI:** 10.3390/nano12132161

**Published:** 2022-06-23

**Authors:** Ayaka Harada, Hiroyasu Tsutsuki, Tianli Zhang, Kinnosuke Yahiro, Tomohiro Sawa, Takuro Niidome

**Affiliations:** 1Faculty of Advanced Science and Technology, Kumamoto University, 2-39-1 Kurokami, Chuo-ku, Kumamoto 860-8555, Japan; 201d8604@st.kumamoto-u.ac.jp; 2Department of Microbiology, Graduate School of Medical Sciences, Kumamoto University, 1-1-1 Honjo, Chuo-ku, Kumamoto 860-8556, Japan; tsutsuki@kumamoto-u.ac.jp (H.T.); zhangtianli220@hotmail.com (T.Z.); sawat@kumamoto-u.ac.jp (T.S.); 3Department of Microbiology and Infection Control Sciences, Kyoto Pharmaceutical University, 5 Misasagi-Nakauchi-cho, Yamashina-ku, Kyoto 607-8414, Japan; kin816tas@mb.kyoto-phu.ac.jp

**Keywords:** PLGA nanoparticles, anti-inflammatory, controlled drug delivery

## Abstract

Advances in drug delivery systems (DDSs) have enabled the specific delivery of drugs to target cells. Subtilase cytotoxin (SubAB) produced by certain enterohemorrhagic *Escherichia coli* strains induces endoplasmic reticulum (ER) stress and suppresses nitric oxide generation in macrophages. We previously reported that modification of SubAB with poly(D,L-lactide-co-glycolic) acid (PLGA) nanoparticles (SubAB-PLGA NPs) increased intracellular uptake of SubAB and had an anti-inflammatory effect on macrophages. However, specific delivery of SubAB to macrophages could not be achieved because its effects on other cell types were not negligible. Therefore, to suppress non-specific SubAB binding, we used low-binding mutant SubAB_S35A_ (S35A) in which the 35th serine of the B subunit was mutated to alanine. In a macrophage cell line, PLGA NPs modified with S35A (S35A-PLGA NPs) induced ER stress and had anti-inflammatory effects similar to WT-PLGA NPs. However, in an epithelial cell line, S35A-PLGA NPs induced lower ER stress than WT-PLGA NPs. These results suggest that S35A is selectively delivered to macrophages rather than epithelial cells by modification with PLGA NPs and exerts anti-inflammatory effects. Our findings provide a useful technique for protein delivery to macrophages and encourage medical applications of DDSs for the treatment of inflammatory diseases.

## 1. Introduction

A drug delivery system (DDS) is an engineered technology for targeted delivery and/or controlled release of therapeutic agents, which plays an important role in advances in pharmaceutical and medical fields. The goal of a DDS is to achieve better therapeutic effects without any side effects. Therefore, the development of attractive drug carriers with both targeting of lesions and controlled release is eagerly awaited [1]. Synthetic nanoparticles (NPs) as drug carriers are a potential clinical therapeutic tool against cancer and inflammation [2]. NPs have been developed on the basis of various materials including liposomes [3], chitosan [4], dextran [5], silica [6], metals such as silver and gold [7], and biodegradable poly(D,L-lactide-co-glycolic) acid (PLGA) [8]. Recently, Jin et al. reported the chirality-controlled protein NPs driven by molecular interactions for cancer therapy. The developed NPs were effectively taken up into HCT116 cells and showed significant antitumor activity [9]. Other polymer materials including hydrogels, micro/nanofibers, and inorganic hybrid NPs have also been developed and are expected to be next-generation DDSs. [10,11,12] Moreover, drugs are delivered to not only specific tissues and cells, but also to specific cellular organelles, such as nuclei, mitochondria, and lysosomes, using a DDS [13,14,15].

The endoplasmic reticulum (ER) is a crucial organelle for protein quality control (e.g., protein synthesis, folding, maturation, and stabilization) [16]. Many intracellular proteins are folded in the ER by various chaperones and folding enzymes [17]. Subsequently, the proteins are subjected to post-translational modifications, including glycosylation and disulfide bond formation by glycosylation enzymes and oxidoreductase, respectively, to generate mature forms [18]. Genetic and environmental damage, such as nutrient depletion, and oxidative stress, causes the accumulation of unfolded proteins, resulting in ER stress. This stress activates the unfolded protein response (UPR) via ER stress sensor proteins [19]. Although ER stress is associated with many pathological conditions, recent studies have revealed that adaptive UPR signaling protects cells from cytotoxicity and excessive inflammation in a context-dependent manner [20]. Therefore, controlling the UPR may provide new prospects for future treatments of ER-related diseases and inflammation. On the basis of the above concept, some researchers have reported targeting the ER using sulfonyl ligands and peptides [21,22]. 

Subtilase cytotoxin (SubAB), a member of the AB_5_ toxin family, was identified in enterohemorrhagic *Escherichia coli* O113:H21 that was associated with an outbreak in Australia [23,24,25]. SubAB consists of an enzymatically active A subunit (SubA) that cleaves host ER chaperon binding immunoglobulin protein (BiP) and a pentameric B subunit (SubB) that binds to sialoglycan-modified cell surface receptors and mediates uptake into target cells [26,27]. SubAB translocates to the ER via the COPI-dependent Golgi retrograde trafficking pathway [28,29]. After translocation into the ER, SubAB cleaves BiP and results in ER stress-induced cytotoxicity through activation of ER stress sensors protein kinase R-like endoplasmic reticulum kinase (PERK), inositol-requiring enzyme 1α (IRE1α), and activating transcription factor 6 (ATF6) that mediate a unique signaling pathway including eukaryotic initiation factor 2α (eIF2α) phosphorylation and C/EBP homologous protein (CHOP) expression [30,31]. Several studies have demonstrated that SubAB induces cytotoxicity, stress granule formation [31], inhibition of autophagy [32] and inflammasome [33] in vitro, and lethal severe hemorrhagic inflammation and enteropathogenic bacterial infection in mice [34]. Additionally, SubAB inhibits lipopolysaccharide (LPS)-induced nitric oxide (NO) production in macrophages through suppression of nuclear factor kappa-B (NF-κB) activation and subsequent inducible NO synthase (iNOS) expression [35]. SubAB may have therapeutic potential for immunosuppression in macrophages because of inhibiting activation of NF-κB, the master regulator of LPS-induced pro-inflammatory gene expression.

Recently, we reported that modification of NPs with PLGA improves the efficiency of wildtype SubAB (WT) uptake into macrophages and the subsequent anti-inflammatory effect [36]. However, macrophage-specific delivery of WT was difficult because the B subunit of WT has a potent cell adhesion property without cell specificity. Therefore, in this study, we attempted macrophage-specific delivery of the toxin using a mutant SubAB with a low-affinity cell adhesion property engineered through site-directed mutagenesis in the B subunit.

## 2. Materials and Methods

### 2.1. Materials

Poly(D,L-lactide-co-glycolic) acid (PLGA, 85:15, MW: 190,000–240,000) and fluorescein isothiocyanate isomer I were purchased from Sigma-Aldrich (St. Louis, MO, USA). N-hydroxysuccinimide (NHS), 1-ethyl-3-(3-dimethylaminopropyl) carbodiimide hydrochloride (water-soluble carbodiimide, WSC), and N-(5-amino-1-carboxypentyl) iminodiacetic acid (AB-NTA) were from DOJINDO (Kumamoto, Japan). A purified mouse anti-BiP/GRP78 antibody (Cat#610979) was from BD Transduction Laboratories™ (Franklin Lakes, NJ, USA). Oleylamine and an anti-β-actin mouse monoclonal antibody were purchased from FUJIFILM Wako Pure Chemical (Osaka, Japan). Anti-mouse immunoglobulin G (IgG), horseradish peroxidase (HRP)-linked (#7076) and anti-rabbit IgG, HRP-linked (#7074), anti-CHOP (#2895), anti-poly(ADP-ribose) polymerases (PARP) (#9542), anti-cleaved PARP (#5625), anti-cleaved cysteine aspartate-specific protease (caspase) 7 (#9491), anti-cleaved caspase 3 (#9661), anti-phospho-eIF2α (#3398), and anti-total eIF2α (#5324) antibodies were obtained from Cell Signaling Technology (Danvers, MA, USA). An anti-iNOS mouse monoclonal antibody [37] and anti-SubAB anti-serum [38] were prepared as reported previously.

### 2.2. Purification of His-Tagged Toxins 

Oligo histidine-tagged (6× His-tagged) recombinant SubAB (His-tagged SubAB; WT), catalytically inactive mutant His-tagged SubA_S272A_B (S272A), and His-tagged A subunit (SubA) were purified by affinity chromatography using Ni-NTA agarose (Qiagen, Hilden, Germany) as reported previously [39].

### 2.3. Site-Directed Mutagenesis of B Subunit of SubAB 

To replace a serine residue (S35) with alanine in the B subunit of SubAB, we performed PCR-based site-directed mutagenesis on a SubAB expression plasmid (pET23b-SubAB) using a QuikChange site-directed mutagenesis kit (Agilent Technologies, Santa Clara, CA, USA), in accordance with the manufacturer’s instructions. Primers used for mutagenesis were as follows: 5′-GGCATGTTTGCAGGCGTTGTTATTACCC-3′ and 5′-CAACGCCTGCAAACATGCCATCCCGGGC-3′ (mutated bases are underlined). The sequence was confirmed by an ABI PRISM 377 DNA sequencer (Applied Biosystems, Foster City, CA, USA). The his-tagged SubAB_S35A_ (S35A) mutant was purified using the same method described above.

### 2.4. HiLyte Fluor™ 555 (HF555) Labeling of Toxins 

To investigate SubAB uptake by RAW264.7 and HeLa cells, WT and mutant toxins (S272A, SubA, and S35A) were labeled with HiLyte Fluor™ 555 Labeling Kit-NH_2_ (Dojindo, Kumamoto, Japan) in accordance with the manufacturer’s instructions. Briefly, to replace the solvent in which the toxins were dissolved, 200 μg toxins were applied to the ultrafiltration spin column and centrifuged at 8000× *g* for 10 min. Toxins were dissolved in reaction buffer, mixed with the succinimidyl ester (SE) reactive form of Hilyte Fluor™ 555 (HF555-SE) (dissolved in DMSO), and incubated at 37 °C for 30 min. WS buffer was added to the mixtures in the spin column and then washed by centrifugation at 8000× *g* for 10 min to remove the unbound HF555-SE. The HF555-labeled toxins (HF555-toxins) were dissolved in 200 µL PBS by pipetting and stored at 4 °C. 

### 2.5. Cell Culture 

Murine macrophage cell lines J774.1 and RAW264.7 cells, and human cervical cancer cell line HeLa were cultured in Dulbecco’s modified Eagle’s medium (FUJIFILM Wako) supplemented with 10% heat-inactivated fetal bovine serum (MP Biomedicals, Santa Ana, CA, USA) and 1% penicillin-streptomycin (Nacalai Tesque, Kyoto, Japan) in a 5% CO_2_ humidified incubator at 37 °C.

### 2.6. Fluorescence Microscopy 

RAW264.7 and HeLa cells were seeded on a glass-bottom dish at 5 × 10^4^ cells/dish and 1×10^4^ cells/dish, respectively. After overnight culture, 100 µL HF555-toxins (HF555-WT, HF555-S272A, HF555-SubA, and HF555-S35A) and 1 µL Cellstain^®^ DAPI solution (DOJINDO) were added to the cells. After incubation for 1 h at 37 °C, cells were observed under a BZ-X800 All-in-one Fluorescence Microscope (Keyence, Osaka, Japan).

### 2.7. Preparation of PLGA NPs 

PLGA NPs were synthesized in accordance with our previous study [36]. Briefly, PLGA and stearic acid at a 1:37 ratio were prepared in chloroform (oil phase). To prepare PLGA NPs, the oil phase was added dropwise to the ultrapure water under probe sonication and evaporation solvent for 3 h.

### 2.8. Preparation of FITC-Conjugated Oleylamine 

FITC-conjugated oleylamine was synthesized by a reaction between the amino group of oleylamine (OA)-fluorescein isothiocyanate isomer I (FITC) and the isothiocyanate group of FITC. FITC (2 mg) was dissolved in 2 mL N,N-dimethylformamide (DMF). Then, 30 mL OA was dissolved in 50 mL DMF. The FITC and OA solution was then mixed for 48 h at 50 °C. After the reaction, FITC-OA was collected by column chromatography.

### 2.9. Preparation of FITC-Labeled PLGA NPs 

FITC-labeled PLGA NPs were prepared by an emulsion/evaporation method. Briefly, PLGA, stearic acid, and FITC-OA at a 1:7:30 ratio were prepared in chloroform (oil phase). To prepare PLGA NPs, the oil phase was added dropwise to the ultrapure water under probe sonication and evaporation solvent for 3 h.

### 2.10. Characterization of PLGA NPs 

The size distribution and zeta potential of NPs were measured by a Zetasizer Nano ZS (Malvern Instruments Ltd., Worcestershire, UK). The shape and size of NPs were observed by transmission electron microscopy (TEM, JEM-1400Plus; JEOL, Tokyo, Japan). For TEM observation, 100 µL PLGA NPs was mixed with 100 µL of 0.01% phosphotungstic acid for 5 min and then 20 µL of the solution was placed on parafilm. A carbon-film-coated TEM grid (ELS-C10, Okenshoji Co., Ltd., Tokyo, Japan) was placed inside the droplet. After incubation for 2 min, the grid was dried in a vacuum at room temperature overnight.

### 2.11. Surface Modification of PLGA NPs with SubAB Toxins 

Modification of the PLGA NP surface with SubAB was performed in accordance with a previous report [30]. In brief, 100 µL PLGA NPs dispersed in water was centrifuged at 12,000× *g* for 5 min. The precipitate was redispersed in 100 µL of a solution containing 10 µL of 0.9 mg/mL NHS and 1.5 mg/mL WSC in 20 mM phosphate buffer (pH 8.2). The mixture was shaken for 20 min at 37 °C, followed by washing with 20 mM phosphate buffer (pH 8.2). Then, 5 µL of 2 mg/mL AB-NTA was added to the NHS-activated PLGA NP solution and the mixture was shaken for 2 h at 37 °C. After washing the mixture, the resultant NTA-modified PLGA NPs were reacted with 5 µL of a 2 mg/mL NiCl_2_ aqueous solution and shaken at 37 °C for 30 min. The mixture was washed with 20 mM phosphate buffer (pH 8.2) and redispersed in 95 µL of 20 mM phosphate buffer (pH 8.2). Then, 5 µL of 347 µg/mL His-tagged SubAB was added. The mixture was shaken for 1 h at room temperature and washed with 20 mM phosphate buffer (pH 8.2). Then, the resultant His-tagged SubAB-conjugated PLGA NPs (WT-PLGA NPs) were stored at 4 °C. Mutant SubAB_S35A_-conjugated PLGA NPs (S35A-PLGA NPs) and SubA-conjugated PLGA NPs (A-PLGA NPs) were prepared in a similar manner.

### 2.12. Evaluation of pH-Dependent Release of His-Tagged SubAB Toxins 

One-hundred microliters of SubAB toxins-PLGA NPs (WT-PLGA NPs, S35A-PLGA NPs, and A-PLGA NPs) were centrifuged (12,000× *g*, 5 min) to remove the supernatant. The NPs were redispersed in 100 µL ultrapure water. The solution was centrifuged again (12000× *g*, 5 min) to remove the supernatant. The NPs were then redispersed in 30 µL of 8 mM phosphate buffer (pH 7.4) or acetate buffer (pH 5.0) and shaken at 37 °C. After 0, 1, 10, and 30 min, the samples were centrifuged (12,000× g, 5 min) to collect the supernatants. Released His-tagged toxins in the supernatants were analyzed by Western blotting using anti-SubAB anti-serum. 

### 2.13. Intracellular Uptake of SubAB Toxin-PLGA NPs

Intracellular uptake of SubAB toxins-PLGA NPs in RAW264.7 and HeLa cells was evaluated by fluorescence microscopy and a fluorescence microplate reader. RAW264.7 and HeLa cells were seeded at 5 × 10^4^ and 1 × 10^4^ cells/dish in glass-bottom dishes at 5 × 10^4^ and 1 × 10^4^ cells/well in a 96-well plate, respectively, and incubated overnight at 37 °C. Cells were incubated for 20 h with FITC-labeled PLGA NPs modified with HF555-labeled SubAB toxins. The distributions of HF555-labeled SubAB toxins and FITC-labeled PLGA NPs were observed under the BZ-X800 All-in-one Fluorescence Microscope. The fluorescence intensity of FITC-labeled PLGA NPs in each well was measured and quantified using a TECAN Infinite F200 Pro fluorescence microplate reader (Tecan Group Ltd., Männedorf, Switzerland) at an excitation wavelength of 485 nm and emission wavelength of 535 nm.

### 2.14. Western Blotting

J774.1 and HeLa cells were seeded at 5 × 10^4^ and 1 × 10^4^ cells/well, respectively, and cultured overnight at 37 °C. The cells were treated with or without LPS (100 ng/mL) in the presence of 5 µg/mL SubAB toxins and 20 µg/mL SubAB toxins-PLGA NPs, and then incubated for 3, 8, or 20 h at 37 °C. Cells were lysed in SDS sample buffer (62.5 mM Tris-HCl, pH 6.8, 2% SDS, 6% glycerol, 0.005% bromophenol blue, and 2.5% 2-mercaptoethanol) and then boiled for 3 min. Proteins were separated by SDS-PAGE, and transferred onto a polyvinylidene difluoride membrane (Merck Millipore, Darmstadt, Germany) at 100 V for 1 h. The membranes were blocked with 5% dry non-fat milk in TBS-T (20 mM Tris-HCl, pH 7.5, 137 mM NaCl, and 0.1% Tween 20) for 1 h and then incubated for 1 h at room temperature or overnight at 4 °C with the indicated primary antibody. The membranes were washed with TBS-T, followed by incubation with an HRP-conjugated secondary antibody at room temperature for 1 h. Protein bands were detected using Immobilon Western Chemiluminescent HRP Substrate (Merck Millipore) and a luminescent image analyzer, the ChemiDoc™ XRS system (Bio-Rad, Hercules, CA, USA).

### 2.15. MTT Assay

Cell viability was assessed by the MTT assay in accordance with our previous study [40]. To evaluate the toxicity of SubAB toxin-PLGA NPs, J774.1 and HeLa cells were seeded at 5 × 10^4^ and 1 × 10^4^ cells/well, respectively, in a 96-well plate and treated for 24 h with or without 5 μg/mL PLGA NPs, WT-PLGA NPs, A-PLGA NPs, or S35A-PLGA NPs. Culture supernatants were then replaced with a culture medium containing 0.75 mg/mL MTT. After 1 h of incubation, a stop solution (isopropanol containing 0.4% HCl and 10% Triton X-100) was added to each well. Relative cell viability was analyzed by measuring absorbance at 570 nm using the microplate reader. Results are expressed as the mean cell viability ± standard deviation (S.D.) of triplicate cultures.

### 2.16. Griess Assay 

Quantification of nitrite ions produced by macrophages was examined by the Griess assay. J774.1 cells seeded in a 96-well plate were treated with or without 5 μg/mL PLGA NPs, WT-PLGA NPs, A-PLGA NPs, or S35A-PLGA NPs. After treatment for 20 h, 50 μL of culture supernatant was mixed with 25 μL Griess Reagent I (1% sulfanilamide in 5% HCl) and incubated for 5 min. Then, 25 μL Griess Reagent II [0.1% N-(1-naphthyl)-ethylenediamine] was added, followed by incubation at room temperature for 10 min. Absorbance at 570 nm was measured by the microplate reader.

### 2.17. Statistical Analysis 

All data are expressed as means ± S.D. Data for each experiment were obtained from at least three independent experiments. Statistical analyses were performed using Student’s *t*-test with the level of significance set at *p* < 0.05.

## 3. Results and Discussion

### 3.1. Cell Recognition-Inactivated SubAB_S35A_ (S35A) Mutant Had Decreased Uptake by Cells

Wildtype SubAB (WT) binds to cell surface receptors and enters through lipid rafts and actin-dependent macropinocytosis-like mechanisms in HeLa cells [26,41,42]. The 35th serine residue (S35) of the B subunit (corresponds to S12 of the secreted B subunit without the N-terminal signal peptide) is required for binding to N-glycolylneuraminic acid-terminated glycans of receptors [23,41]. To evaluate cellular uptake of mutant SubAB toxins including the catalytically inactive mutant SubA_S272A_B (S272A), site-directed B subunit mutant SubAB_S35A_ (S35A), and B subunit-deficient mutant SubA, we treated cells with fluorescent dye-labeled toxins [HF555-wildtype SubAB (WT), S272A, S35A, and SubA (A)] and observed the distribution by fluorescence microscopy. Figure 1 shows fluorescent signals of HF555 around nuclei in both RAW264.7 and HeLa cells treated with HF555-WT and HF555-S272A, suggesting that both WT and mutant S272A, which have intact B subunits, were internalized and localized in the ER (Figure 1a,b). However, intracellular accumulation of HF555 fluorescence was not observed in cells treated with HF555-A and HF555-S35A. In agreement with a previous report [23,43], our results indicated that intracellular uptake of SubAB was dramatically decreased by deficiency of the B subunit or mutation in serine at position 35. *Yersinia pestis* and *Salmonella enterica* serovar Typhi encode homologs of SubB, namely YpeB and PltB [44,45]. Mutations in the serine residue corresponding to S35 of SubB negate the binding activity, demonstrating the importance of the serine residue at this position for sialoglycan-modified cell surface receptors. However, mutation in the active site of the A subunit did not affect cellular uptake and accumulation around the nucleus. 

SubAB is known to specifically cleave the endoplasmic reticulum chaperone BiP. Figure 1c shows the Western blot images of BiP cleavage by incubation of RAW264.7, HeLa, and HeLa cell lysates with the indicated toxins. WT cleaved BiP in both RAW 264.7 and HeLa cells, and in cell lysates, whereas SubA (A) and SubAB_S35A_ (S35A) cleaved BiP only in cell lysates (Appendix A). These results indicate that both A and S35A have an enzymatic activity that cleaves BiP but cannot enter cells because of reduced cell binding. Therefore, the accumulation of HF555-SubAB toxins around the nucleus indicates ER accumulation that depends on the binding activity of the B subunit. However, S272A translocates to the ER but does not cleave BiP in macrophages, HeLa cells, or cell lysates because of mutation in the active site.

### 3.2. Preparation and Characterization of PLGA NPs Modified with Various SubAB Toxins

Poly(D,L-lactide-co-glycolic) acid (PLGA) is a synthetic copolymer composed of lactic and glycolic acids. PLGA has amphipathic properties because of the lactic acid-derived hydrophobic crystalline characteristic and glycolic acid-derived hydrophilic amorphous characteristic. It is degraded by cellular hydrolysis and metabolized by the tricarboxylic acid cycle, and subsequently eliminated from the body as carbon dioxide and water [46]. PLGA NPs are one of the most characterized biomaterials available for the development of drug delivery systems in terms of design and performance as a biodegradable micro/nano-device [47,48]. To expose carboxyl groups outside of the particle, we prepared PLGA NPs by the single oil-in-water emulsion method in presence of stearic acids (Figure 2a). TEM observation showed that the PLGA NPs were spherical and dispersed individually (Figure 2b). The PLGA NPs had an average particle size of 260.7 ± 14.7 nm and a zeta potential of −32.8 ± 0.32 mV (Table 1 and Appendix A). The observed size of the PLGA NPs on the TEM grids was smaller than the diameters measured by DLS, which is related to the different states of the particles during these measurements. In DLS measurement, the NPs were dispersed in an aqueous solution and therefore swollen [49,50]. Meanwhile, in TEM observation, the NPs were observed in dry conditions. Thus, the NPs may be shrunk during the dry process, resulting in a smaller size than that of DLS measurement [51]. Furthermore, DLS measurement is a technique to characterize particle size from the decay of light scattering as a result of Brownian motion. In the case of a high polydispersity index, light scattering intensity is strengthened by large particles, meaning that the size of characterized particle was larger than that of the intrinsic particle [52]. The negative zeta potential indicated that the carboxyl groups of stearic acid were embedded in the NP surface. The carboxyl groups of stearic acid were modified by His-tagged WT, A, or S35A through a Ni^2+^ chelate complex (Figure 2c). The poly-histidine-tagged (His-tagged) recombinant protein binds strongly to transition metal chelates such as the Ni(II) nitrilotriacetate (Ni-NTA) complex at pH 8.0. Protonation of the imidazole nitrogen atom in the histidine residue (pKa_3_ = 6.04) and the coordination bond between histidine and transition metal ions including Ni^2+^ are disrupted by a pH reduction to 5.0 [53,54]. In a previous study, macrophage-selective delivery was not achieved by WT-PLGA NPs. WT-PLGA has also delivered WT to epithelial cells. Thus, to target macrophages, we modified PLGA NPs through the Ni-NTA complex at pH 8.0 with the His-tagged A and S35A, which has a low cell-binding activity. The particle size and zeta potential of WT-, A-, and S35A-PLGA NPs were examined by DLS. As shown in Table 1 and Appendix A, the average size of the toxin-modified PLGA NPs was approximately 300 nm, and the zeta potential had increased positively. Theoretically calculated isoelectric point (pI) values of SubAB subunits obtained by the Compute pI/Mw tool ExPASy (https://web.expasy.org/computepi/) (accessed on 14 February 2022) were 9.28 (A subunit) and 8.54 (B subunit) in water. These results suggested that PLGA NPs were successfully modified with toxins via the stearic-acid-based Ni-NTA complex. 

To evaluate pH-responsive dissociation of His-tagged SubAB toxins and SubA from PLGA NPs, we treated WT-, A-, and S35A-PLGA NPs with buffer solutions of different pH values as indicated in Appendix A. Most His-tagged toxins had dissociated within 1 min at pH 5.0, but not at pH 7.4 (Appendix A). Recently, functional vesicular systems consisting of non-ionic surfactants and cholesterol such as niosome have attracted much attention in the field of drug delivery because of their advantages, such as being capable of entrapping both hydrophilic and hydrophobic drugs in their aqueous inner core and lipid bilayer, high stability, biocompatible, biodegradable [55,56]. Niosome is a well-designed bilayer membrane carrier, but it takes several hours to release the drug [57]. On the other hand, the release of toxins from PLGA NPs is considerably faster and is expected to have a rapid drug effect in an acidic environment. Consistent with a previous study [36], these data suggested that the His-tagged toxins would be released from PLGA NPs in a pH-dependent manner when taken up by cells and exposed to the acidic environment of endosomes and lysosomes. This study evaluated the stability of NPs in water (Table 1 and Appendix A) and the release of toxins under acid conditions (Appendix A). However, in a biological environment containing blood or plasma, the stability of NPs may change and uncontrolled toxin release may occur. Therefore, further studies under various biological conditions are needed to better explore its stability and toxin release.

### 3.3. WT- and S35A-PLGA NPs Induce ER Stress and Inhibit iNOS Expression in Macrophages

SubAB cleaves BiP to produce 44 and 28 kDa polypeptides [27]. The cleavage of BiP induces an ER stress response, leading to expression of CHOP, cleavage of PARP, and activation of caspases [58,59,60]. To investigate the effect of toxin-modified PLGA NPs on J774.1 cells, the cells were incubated for 3, 8, and 20 h with free or toxin-modified PLGA NPs, followed by detection of BiP cleavage, PARP cleavage, and CHOP expression.

After 3 h of incubation, free WT cleaved BiP, but not free S35A, A, and PLGA NPs alone. However, WT and S35A-modified PLGA NPs induced BiP cleavage, but not A-PLGA NPs (Figure 3a,b). PARP was cleaved from 116 kDa to 85 kDa as apoptosis progressed. Evaluation of PARP cleavage at 8 h revealed more fragments in cells treated with WT, WT-PLGA NPs, and S35A-PLGA NPs compared with other treatments (Figure 3c,d). Consistent with the results of BiP and PARP cleavage, CHOP expression at 8 and 20 h was higher after WT, WT-PLGA NP, and S35A-PLGA NP treatments compared with other treatments. S35A-PLGA NPs induced a lower level of CHOP expression compared with WT at 8 h, but the expression was equal to WT at 20 h (Figure 3c,e,f and Appendix A). In any time period, A did not induce PARP cleavage or CHOP expression even when modified to PLGA NPs. Therefore, these results suggest that S35A-PLGA NPs affect macrophages, although it is slightly less than WT and WT-PLGA NPs.

Previously, we reported that SubAB-mediated ER stress inhibits LPS-induced iNOS expression by suppressing nuclear translocation of NF-κB [35]. iNOS is an inflammatory marker, which produces NO from L-arginine as a substrate [61]. The produced NO is oxidized to nitrite as a stable metabolite [62]. At 20 h of treatment, iNOS and nitrite generation were quantified by Western blotting and the Griess assay, respectively. S35A-PLGA NPs inhibited iNOS expression and nitrite production in addition to free WT and WT-PLGA NPs (Figure 3g–i). Additionally, cell viability assessed by MTT assays after treatment for 24 h revealed that WT, WT-PLGA NPs, and S35A-PLGA NPs showed cytotoxicity (Appendix A). 

These results revealed that free S35A did not induce ER stress and suppressed inflammation because it was not taken up by cells. The PLGA NP modification induced S35A uptake by J774.1 cells, followed by ER stress due to BiP cleavage, and suppressed iNOS expression and nitrite production in J774.1 cells. However, A-PLGA NPs did not induce ER stress or BiP cleavage, suggesting that the B subunit of SubAB is important not only for cell adhesion but also for delivery of the A subunit to the ER. In general, the ER retention signal sequence, such as KDEL and KKXX-like motif sequences, play a crucial role in the localization of transmembrane proteins in the ER [63], and the B subunit has the KKNS sequence at amino acids 128–131. Although S35A has a mutation in serine at position 35 of the B subunit to alanine, this mutation does not affect the sequence of KKNS. Therefore, WT and S35A transferred to the ER via KKNS sequences after being taken up by macrophages.

### 3.4. S35A-PLGA NPs Do Not Induce ER Stress or Cytotoxicity in HeLa Cells

As shown in Figure 3, S35A-PLGA NPs showed anti-inflammatory effects in macrophages. To develop macrophage-specific delivery, it was necessary to prove that S35A-PLGA NPs did not enter non-phagocytic cells such as epithelial cells and do not induce ER stress. Therefore, we used HeLa cells to investigate whether S35A-PLGA NPs induced ER stress in non-phagocytic cells. Caspase is activated by the ER stress response and is a useful marker of cell death due to apoptosis [64]. Phosphorylation of eIF2α at Ser51 is induced by various environmental stresses such as ER and oxidative stresses or amino acid starvation. There are four types of protein kinases known to phosphorylate eIF2α in mammals, including PERK, general control nonderepressible-2 (GCN2), heme-regulated inhibitory (HRI), and protein kinase R (PKR) [65]. PERK activation induces eIF2α phosphorylation, caspase 3 and 7 cleavage, CHOP expression, and PARP cleavage [66,67,68,69]. We next evaluated eIF2α phosphorylation, caspase and PARP cleavage, and CHOP expression to assess the cytotoxicity of S35A-PLGA NPs

We evaluated the BiP cleavage, eIF2α phosphorylation, and cleaved caspase 7 at 3 h in HeLa cells (Figure 4a,b). WT and WT-PLGA NPs cleaved BiP, which correlated with caspase 7 activation and eIF2α phosphorylation. Interestingly, BiP cleavage was not detected in S35A-PLGA NP-treated HeLa cells (Figure 4a,b and Appendix A). Caspase 7 activation and eIF2α phosphorylation in S35A-PLGA NP-treated cells were lower than those in WT or WT-PLGA NP-treated cells. At 8 h, PARP cleavage, as well as caspase 3 and 7 activation, were detected in WT or WT-PLGA NP-treated HeLa cells (Figure 4c,d and Appendix A), but not in cells treated with S35A-PLGA NPs. CHOP expression induced by S35A-PLGA NPs was hardly detected compared with WT or WT-PLGA NP treatments, suggesting that ER stress did not occur in HeLa cells (Figure 4e,f). Consistent with these results, WT and WT-PLGA, but not S35A-PLGA NPs, caused cell death after 24 h of treatment (Appendix A). ER stress mediated by activation of the PERK-eIF2α pathway is important as the main cell death mechanism induced by SubAB [70]. SubAB inhibits protein synthesis through eIF2α phosphorylation and caspase 3/7 activation and fragmentation, followed by apoptosis [71,72]. Notably, S35A-PLGA NPs showed lower activation of these factors than WT-PLGA NPs in HeLa cells. By summarizing the results from Figure 3, Figure 4, Appendix A, our data suggested that S35A-PLGA NPs induce ER stress and cytotoxicity in macrophages through activation of these cell death mechanisms, but not in endothelial cells. 

### 3.5. S35A-PLGA NPs Specifically Enter Macrophages, but Not Epithelial Cells

To examine the cellular uptake of PLGA NPs by RAW264.7 and HeLa cells, we synthesized FITC-labeled PLGA NPs and then modified them with WT or S35A. The intracellular localization of these fluorescent PLGA NPs was evaluated by fluorescence microscopy and a fluorescence microplate reader. Fluorescence images revealed that FITC-labeled PLGA NPs were taken up by macrophages regardless of the toxin type (Figure 5a). Additionally, evaluation of PLGA NP uptake by the fluorescence microplate reader revealed no significant difference in the uptake by RAW264.7 cells of any of the NPs (Figure 5b). However, WT-PLGA NPs were taken up and HF555-WT was detected in HeLa cells, but not PLGA NPs alone or S35A-PLGA NPs (Figure 5c). The fluorescence intensity read by the microplate reader showed that S35A-PLGA NPs did not enter HeLa cells (Figure 5d). These data suggested that S35A-PLGA specifically targets macrophages. In addition to macrophages, S35A-PLGA NPs have the potential to target neutrophils and dendritic cells that have a phagocytic activity.

WT-PLGA NPs exert a higher effect on macrophages than free SubAB at low concentrations [36]. It has been considered that this may be due to uptake of PLAGA NPs via phagocytosis caused by the entry of SubAB through other pathways or promotion of the normal pathway, including endocytosis or a macropinocytosis-like mechanism. However, WT-PLGA NPs showed cytotoxicity in HeLa cells (Appendix A), which are non-phagocytic epithelial cells. SubAB enters HeLa cells via an actin/lipid-raft-dependent macropinocytosis-like uptake pathway [26]. The B subunit binds to sialoglycan which mainly has N-glycolylneuraminic acid as a sialic acid [43]. S35A is a synthetic mutant in which the 35th serine (S35) of the B subunit is replaced with alanine, but the BiP cleavage ability of the A subunit and ER retention signal sequence of the B subunit are active. Although the entry of S35A into cells was reduced, if taken up, it was transported to the ER, cleaved BiP (Figure 1c), and induced ER stress in macrophages (Figure 3). These data indicated that S35 is responsible for the binding activity of the B subunit. By focusing on this property, we conceived delivery using the phagocytosis of PLGA NPs by macrophages. In fact, S35A was not incorporated as a free toxin, but S35A-PLGA NPs were taken up by macrophages (Figure 5a), leading to ER stress and anti-inflammatory effects (Figure 3). As expected, S35A-PLGA NPs did not enter HeLa cells (Figure 5b) and produced neither ER stress nor cytotoxicity. Interestingly, we achieved specific delivery of S35A to the macrophage ER by targeting the uptake with PLGA NP. Previous KO cell screening using the CRISPR/Cas9 gRNA library has demonstrated that KDEL receptors 1 and 2 are important host factors for SubAB toxicity [41]. Cell death is attenuated in KDELR1 and 2 KO cells, suggesting that the ER retention KKNS sequence in the C-terminal of the SubAB B subunit contributes to the localization of the ER and the cytotoxicity of SubAB. S35A reached the ER via this sequence, but SubA was unable to suppress macrophage inflammation even when modified with PLGA NPs. As SubA does not have an ER retention KKNS sequence, these results suggest that SubA cannot reach the ER even if this subunit enters the cell with PLGA NPs. Our observation raises the possibility that PLGA NPs modified with SubB containing an ER retention signal sequence could deliver a reagent of interest to the ER of phagocytic cells. By modifying proteins or peptides with organelle-targeting sequences, our system may deliver substances to various organelles such as nuclei and mitochondria. The ER and Golgi pathways play important roles in the post-translational modification and maturation of synthesized proteins. In particular, the ER is an essential organelle for redox regulation of disulfide bond formation and sugar chain modification, and enzymes that mediate these maturation processes are mainly abundant in the ER. In the future, it is expected that this technology will be applied to DDSs that target organelles, leading to the development of treatment strategies for many diseases such as infectious diseases and cancer as well as inflammation.

## 4. Conclusions

In this study, we established a macrophage-specific delivery system using site-directed mutagenesis of bacterial toxins and PLGA NP modification. SubAB_S35A_ (S35A), which has a mutated serine residue (S35) of the B subunit to reduce cell-binding activity [23,43], was not taken up by cells without the PLGA modification. However, S35A-PLGA NPs were efficiently taken up by J774.1 cells and suppressed inflammation. S35A-PLGA NPs were hardly taken up by HeLa cells and did not show toxicity, suggesting that S35A PLGA NPs do not have a non-specific effect on epithelial cells. Additionally, PLGA NPs conjugated with His-tagged S35A via the Ni-NTA complex resulted in rapid dissociation of S35A from PLGA NPs at a low pH, which was similar to the results observed in our previous study. This suggests that S35A is efficiently released from NPs by intracellular acidification during cellular trafficking after phagocytosis. Our well-designed DDS has achieved macrophage-specific delivery of anti-inflammatory toxins and it is exerting their effects in the ER. S35A-PLGA NPs are nano-sized materials that are biodegradable and expected to be an anti-inflammatory nanomaterial with low side effects. However, further confirmation of the stability of PLGA NPs and the release of toxins from NPs under biological conditions is necessary in future studies. Toxins from various microorganisms are bioactive and act specifically on the target molecule in the host. Many toxins have useful functions, such as anti-inflammatory, anti-cancer, and immune-potentiating effects, to support bacterial survival and transmission. Therefore, such bioactive toxins could be applied as new therapeutic agents without side effects if their binding activity and cellular uptake are regulated by modification with NP. These results are also expected to be applied to organelle-specific delivery of drugs and therapeutic strategies for autoimmune diseases, infection, and cancer. This study has been limited to experiments using cultured cells, especially mouse macrophages. In the future, we would like to establish a therapeutic treatment for mouse inflammation models and an experimental system using human NO-producing cells and a co-culture system consisting of macrophages and epithelial cells, and evaluate the potential of anti-inflammatory NPs for clinical application.

## Figures and Tables

**Figure 1 nanomaterials-12-02161-f001:**
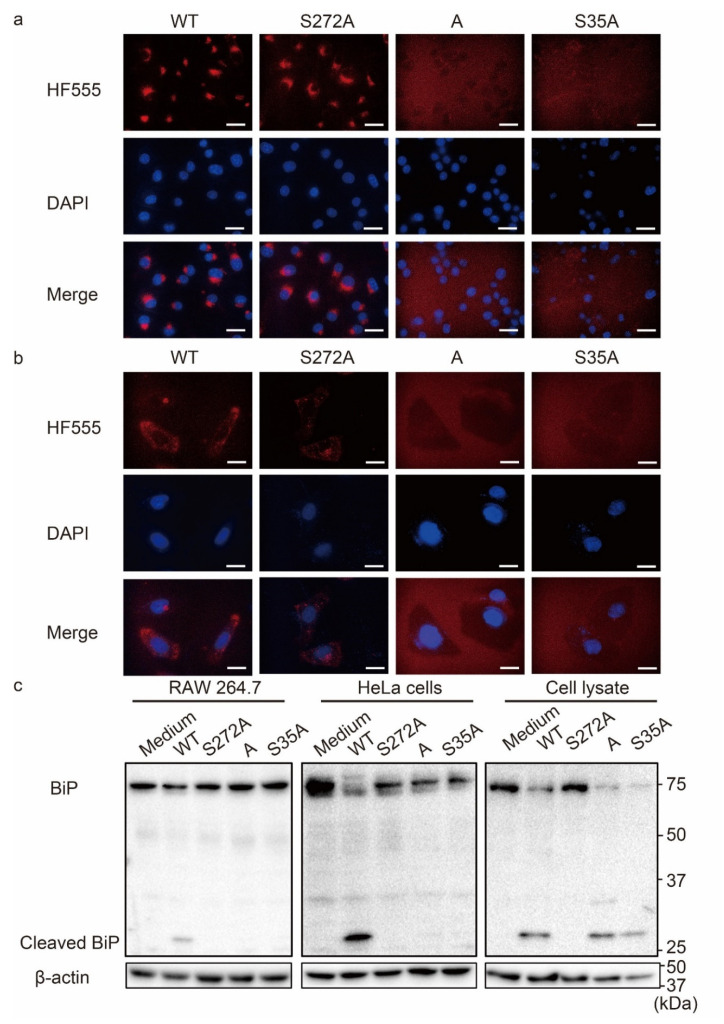
Evaluation of cellular uptake and BiP cleavage activity of various SubAB toxins. (**a**,**b**) RAW264.7 cells (**a**) and HeLa cells (**b**) were treated with 5 µg/mL Hylite555 (HF555)-labeled toxins and then incubated for 1 h at 37 °C. After adding a DAPI solution, cells were observed by fluorescence microscopy. WT: HF555-labeled wildtype SubAB; S272A: HF555-labeled SubA_S272A_B; A: HF555-labeled SubA; S35A: HF555-labeled SubAB_S35A_. Scale bars: 10 µm. (**c**) RAW264.7 and HeLa cells and a HeLa cell lysate were treated for 1 h with 5 µg/mL toxins. BiP cleavage in cultured cells and the cell lysate were evaluated by Western blotting. WT: wildtype SubAB; S272A: SubA_S272A_B; A: SubA; S35A: SubAB_S35A_. β-Actin served as a loading control.

**Figure 2 nanomaterials-12-02161-f002:**
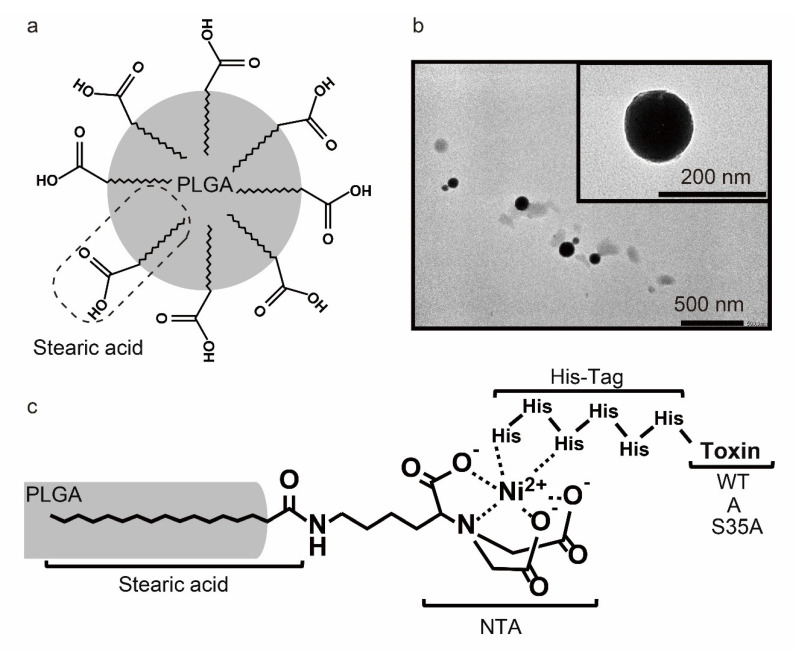
Preparation of PLGA NPs and modification with SubAB. (**a**) Schematic diagram of PLGA NPs in which stearic acids are anchored on the surface. (**b**) Transmission electron microscopy image of PLGA NPs. (**c**) Schematic diagram of toxin-PLGA NPs in which toxins are anchored on the surface.

**Figure 3 nanomaterials-12-02161-f003:**
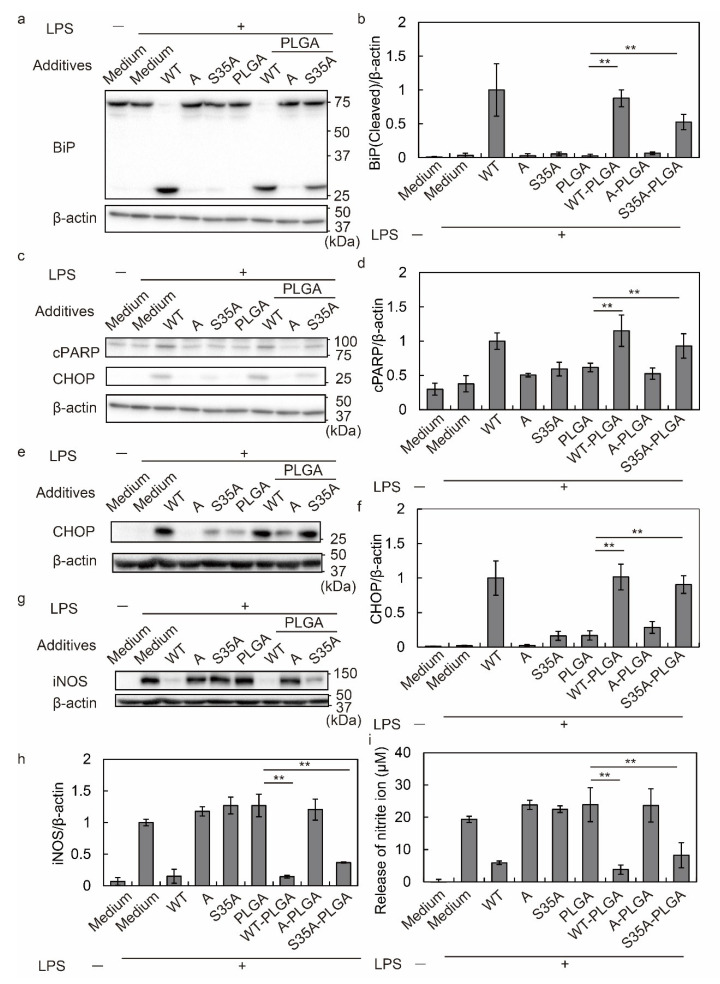
Induction of ER stress and anti-inflammatory effect by SubAB-PLGA NPs on macrophages. (**a**–**h**) J774.1 cells were treated with or without LPS (100 ng/mL) in the presence or absence of 5 µg/mL toxins and 20 µg/mL PLGA NPs for 3, 8, and 20 h. After incubation, the cells were lysed in SDS sample buffer and subjected to Western blotting. (**a**) Evaluation of BiP cleavage at 3 h by Western blot analysis. (**b**) Quantification of BiP cleavage by densitometry. (**c**) Evaluation of cleaved PARP (cPARP) and CHOP expression at 8 h by Western blot analysis. (**d**) Quantification of PARP cleavage by densitometry. (**e**) Evaluation of CHOP expression at 20 h by Western blot analysis. (**f**) Quantification of CHOP expression by densitometry. (**g**) Evaluation of iNOS expression at 20 h by Western blot analysis. (**h**) Quantification of iNOS expression by densitometry. (**i**) Quantification of nitrite production by the Griess assay. Data are expressed as means ± S.D. (*n* = 3). Statistical analyses were performed using Student’s *t*-test. ** *p* < 0.01.

**Figure 4 nanomaterials-12-02161-f004:**
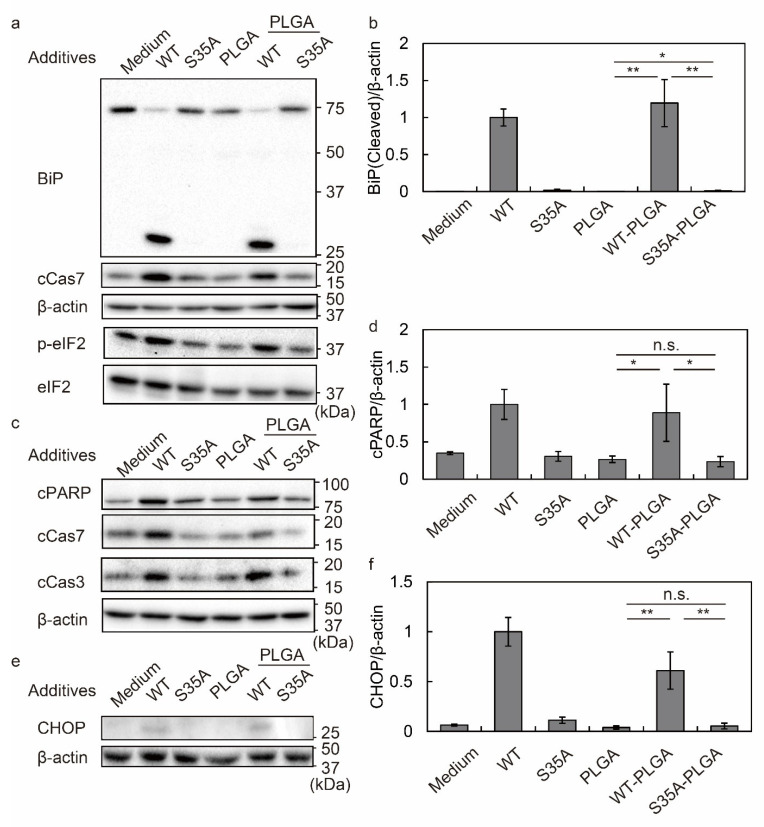
Induction of ER stress and cytotoxicity by SubAB-PLGA NPs in HeLa cells. (**a**–**f**) HeLa cells were treated with or without 5 µg/mL toxins and 20 µg/mL toxin-PLGA NPs for 3, 8, and 20 h. After treatment, the cells were lysed in SDS sample buffer and subjected to Western blotting. cCas7, cleaved caspase 7; cCas3, cleaved caspase 3; cPARP, cleaved PARP. (**a**) Evaluation of BiP cleavage, caspase 7 activation, and eIF2 phosphorylation at 3 h. (**b**) Quantification of BiP cleavage by densitometry. (**c**) Evaluation of PARP cleavage, and caspase 3 and 7 activation at 8 h. (**d**) Quantification of PARP cleavage by densitometry. (**e**) Evaluation of CHOP expression at 20 h. (**f**) Quantification of CHOP expression by densitometry. Data are expressed as means ± S.D. (*n* = 3). Statistical analyses were performed using Student’s *t*-test. n.s., not significant. * *p* < 0.05, ** *p* < 0.01.

**Figure 5 nanomaterials-12-02161-f005:**
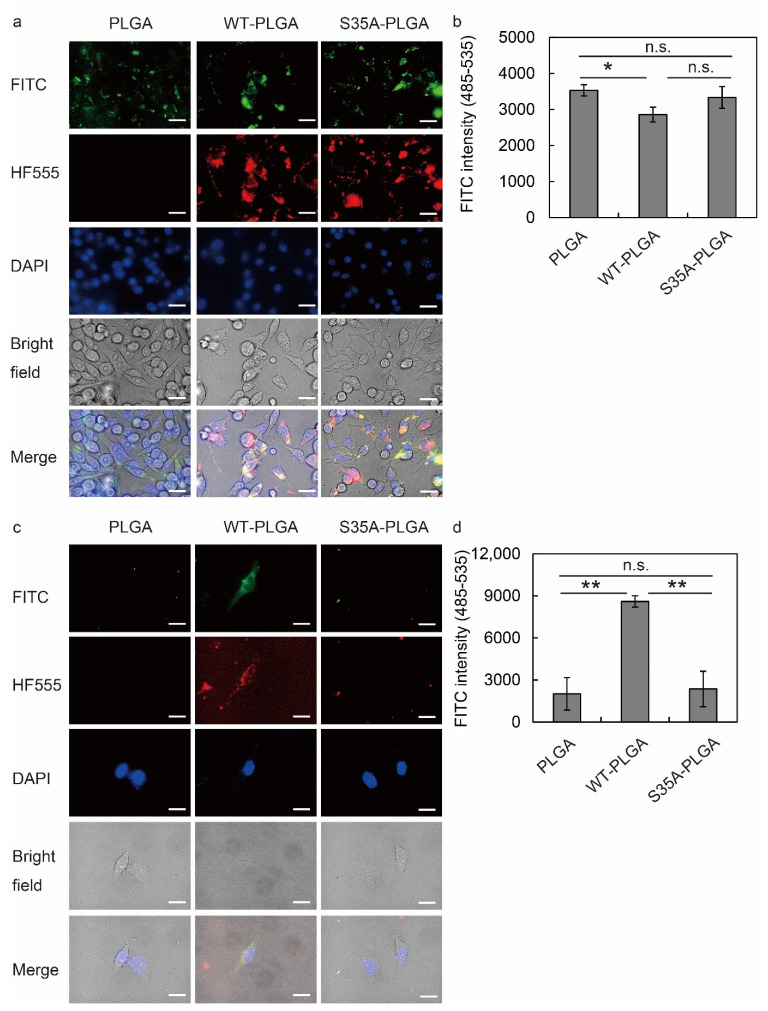
Uptake of SubAB-PLGA NPs by RAW264.7 and HeLa cells. RAW264.7 (**a**,**b**) and HeLa (**c**,**d**) cells were treated with 5 µg/mL FITC-labeled PLGA NPs modified with or without HF555-labeled WT or S35A. After incubation for 20 h, intracellular localization of fluorescent dye-labeled PLGA NPs and toxins was analyzed by fluorescence microscopy (**a**,**c**) and a fluorescence microplate reader (**b**,**d**). Scale bars: 10 µm. Data are expressed as means ± S.D. (*n* = 3). Statistical analyses were performed using Student’s *t*-test. n.s., not significant. * *p* < 0.05, ** *p* < 0.01.

**Table 1 nanomaterials-12-02161-t001:** Particle size, polydispersity index, and zeta potential of PLGA NPs, WT-PLGA NPs, A-PLGA NPs, and S35A-PLGA NPs in water.

	Size (nm)	Polydispersity Index	Zeta Potential (mV)
PLGA NPs	260.7 ± 14.7	0.215 ± 0.012	−32.8 ± 0.32
WT-PLGA NPs	303.4 ± 4.9	0.367 ± 0.030	7.09 ± 1.32
A-PLGA NPs	302.5 ± 5.6	0.355 ±0.008	8.89 ± 0.73
S35A-PLGA NPs	302.7 ± 2.3	0.362 ±0.035	3.19 ± 0.15

Data are expressed as means ± S.D. (*n* = 3).

## Data Availability

The data presented in this study are available on request from the corresponding author.

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
