# Peer review of "Controlled Delivery of an Anti-Inflammatory Toxin to Macrophages by Mutagenesis and Nanoparticle Modification"

_nanomaterials, 2022, doi:10.3390/nano12132161_

Round 1

Reviewer 1 Report

The use of a modified subtilase cytotoxin (SubAB) with PLGA nanoparticles as a biological anti‐inflammatory toward macrophages, is interesting and merits being further explored: in this regard the work is original and sound.

Nothwidstaning, some aspects need to be detailed or justified in more detail, such as the stability of the linked NPs to mutant protein (not only in respect to the chemical stability but the biological stability in plasma and blood), as well as a detailed analysis of data in co-cultures of cells, in order to understand if the toxin is mobile, once released (a free form of the protein) in an acidic condition in presence of cellular damage and permeability. This aspect may be relevant since the variation of the microenvironment and cell infiltration may pave the way to an uncontrolled release of toxins: also, the Authors considered the HeLa model sufficient to demonstrate the safety of the proposed approach? Does the difference between human and murine macrophages impact the translation of this approach toward clinical applications for the treatment of inflammatory diseases?

The manuscript thus requires a major revision to be considered for publication and may require flow diagrams to help the synthesis and legibility.

Author Response

Reviewer Comment. The use of a modified subtilase cytotoxin (SubAB) with PLGA nanoparticles as a biological anti-inflammatory toward macrophages, is interesting and merits being further explored: in this regard the work is original and sound.

Response: We would like to thank this Reviewer for critical and encouraging comments to us. The comments are very helpful for improving this manuscript. For each comment, we responded and revised the original manuscript as follows.

Comment: Nothwidstaning, some aspects need to be detailed or justified in more detail, such as the stability of the linked NPs to mutant protein (not only in respect to the chemical stability but the biological stability in plasma and blood), as well as a detailed analysis of data in co-cultures of cells, in order to understand if the toxin is mobile, once released (a free form of the protein) in an acidic condition in presence of cellular damage and permeability. This aspect may be relevant since the variation of the microenvironment and cell infiltration may pave the way to an uncontrolled release of toxins: also, the Authors considered the HeLa model sufficient to demonstrate the safety of the proposed approach? Does the difference between human and murine macrophages impact the translation of this approach toward clinical applications for the treatment of inflammatory diseases?

Response: We thank the reviewer for the critical comments. As the reviewer pointed out, biological stability would be important for therapeutic application in our future. Unfortunately, fresh plasma and blood from animals were not readily available and could not be verified experimentally. Other groups have reported that PLGA NPs are stable in solutions containing serum (DOI:10.1007/s12035-015-9235-x) and do not cause hemolysis (DOI: 10.1039/c5nr00733j). These reports suggest that our PLGA NPs is also stable in serum.

Although detailed data cannot be provided (because it is important data for other studies), when HeLa cells and macrophages were co-cultured and treated with PLGA NPs, the particles predominantly entered into macrophages. Regarding the cytotoxicity of toxin alone, Figures S4b and S5e show that WT was toxic, while S35A was almost non-toxic. Therefore, our results suggest that PLGA NPs may target macrophages, and that even if the toxin dissociates from PLGA NPs under acidic conditions, S35A alone does not damage the cells.

Since this study has not yet focused on a specific disease, we used HeLa cells, which are common epithelial cells other than macrophages. As this Reviewer kindly suggested, we are wondering if this study could be applied to the treatment of certain inflammatory diseases, and we need to use model cells suitable for the experiment. In the future, we would like to further investigate disease models that should be therapeutic targets and particle administration methods.

Finally, we used mouse macrophages here, but as this reviewer suggested, we believe that human macrophages model is important for clinical application. Previously, we found that SubAB suppresses LPS-induced NO production in mouse macrophages (Tsutsuki et al., 2012). Based on this finding, we have developed biodegradable anti-inflammatory NPs that can efficiently suppress NO production in macrophages by using PLGA NPs as a carrier for SubAB delivery (Harada et al., 2020). In this paper, we developed PLGA NPs using mutant toxins to improve their specificity against macrophages, and evaluated whether they could selectively suppress the NO production in macrophages. We have consistently used mouse macrophage cell line and conducted experiments to assess anti-inflammatory properties by comparison with previous data. We first develop the effective NPs using the mouse macrophages model and evaluate its efficacy and safety in the future by in vivo experiments using inflammatory disease model mice. For clinical application, we would like to evaluate using human macrophages as well. As a preliminary result, we tried to induce NO production using PMA-differentiated THP-1 cells-derived macrophages, which is a generally used as a human macrophage model, but we could not induce NO production. In general, mouse and human have different promoter properties of inducible NO synthase (iNOS) whose expression is induced by various stimuli in macrophages. Therefore, it is controversial whether human cell lines can induce sufficient NO production. While conducting research using a mouse model, we would like to establish an evaluation system using NO-producing human cells. Therefore, we added sentences to revised manuscript Page 16, Lines 512-516 as follows:

“This study has been limited to experiments using cultured cells, especially mouse macrophages. In the future, we would like to establish a therapeutic treatment for mouse inflammation models and an experimental system using human NO-producing cells, and evaluate the potential of anti-inflammatory nanoparticles for clinical application.”

Reviewer 2 Report

The paper tried a macrophage‐specific delivery of the toxin using a mutant SubAB with a low affinity cell adhesion property engineered through site‐directed mutagenesis in B subunit. The paper have some interesting results and can be considered after addressing the following issues:

-please add Anti‐inflammatory to the keywords

-it must add size distribution and zeta potential curves of DLS analysis. It gives a better insight to see a distribution of them.

-why do PLGA NPs have a zeta of -30 mv? What is the nature of this negative zeta?

-there are some other nanocarriers that reported recently, it suggests to discuss them: https://doi.org/10.1016/j.jddst.2020.101916, https://doi.org/10.1021/acs.nanolett.0c01757, https://doi.org/10.1016/j.ajps.2020.05.003, https://doi.org/10.3390/polym14061259

-for other formulations, you report a low positive zeta potential (about +8 mV). Is it a stable formulation?

-recently niosome gained lots of interest for drug delivery, it suggests to compare them with PLGA NPs: https://doi.org/10.1007/s10856-021-06623-6, https://doi.org/10.1021/acsomega.1c03816

-authors used some abbreviations without any explanation, it must be added

-why does DLS have a bigger size than TEM? Can u explain that?

Author Response

Reviewer Comment. The paper tried a macrophage-specific delivery of the toxin using a mutant SubAB with a low affinity cell adhesion property engineered through site-directed mutagenesis in B subunit. The paper have some interesting results and can be considered after addressing the following issues

Response: Thank you very much for reviewing our manuscript and for your very kind and encouraging comments to us. The comments are very helpful for improving this manuscript. For each comment, we responded and revised the original manuscript as follows.

Comment: please add Anti-inflammatory to the keywords

Response: Thank you for your valuable comment. We added Anti-inflammatory to the keywords.

Comment: it must add size distribution and zeta potential curves of DLS analysis. It gives a better insight to see a distribution of them.

Response: We appreciate this Reviewer’s helpful comment. We added size distribution and zeta potential curves of DLS analysis in FigureS2. Please kindly see the revised supporting information.

Comment: why do PLGA NPs have a zeta of -30 mv? What is the nature of this negative zeta?

Response: Thank you for your valuable comment. PLGA is a copolymer of lactic acid and glycolic acid, and has carboxyl groups at the ends of the polymer. In our PLGA NPs, the hydrophobic part of stearic acid anchors the PLGA, and the carboxyl groups are exposed on the particle surface. Therefore, the PLGA NPs have a negative zeta potential due to the carboxyl groups.

Comment: there are some other nanocarriers that reported recently, it suggests to discuss them:https://doi.org/10.1016/j.jddst.2020.101916,https://doi.org/10.1021/acs.nanolett.0c01757,https://doi.org/10.1016/j.ajps.2020.05.003,https://doi.org/10.3390/polym14061259

Response: Thank you for your suggestion. We added that suggested research of protein nanoparticles for cancer therapy in the introduction as "Recently, Jin et al reported the chirality-controlled protein nanoparticles driven by molecular interactions for cancer therapy. The developed nanoparticles were effectively taken up into HCT116 cells and showed significant antitumor activity [9]." in Page 1-2, Line 42-45, please kindly see it.

Comment: for other formulations, you report a low positive zeta potential (about +8 mV). Is it a stable formulation?

Response: Thank you for your valuable comment. The stability of the PLGA NPs modified with toxins slightly lowered compared with the original PLGA NPs due to low zeta potential and increased Polydispersity index. However, we thought that the PLGA NPs modified with toxins were still stable in the physiological environment because of the no aggregation of NPs during experiments.

Comment: recently niosome gained lots of interest for drug delivery, it suggests to compare them with PLGA NPs:

https://doi.org/10.1007/s10856-021-06623-6, https://doi.org/10.1021/acsomega.1c03816

Response: We appreciate this Reviewer’s helpful comment. In response to this comment, we compared our PLGA NPs with niosome and added following sentence to the revised manuscript as that in the result and discussion as "Recently, functional vesicular systems consisted of non-ionic surfactants and cholesterol such as niosome have attracted much attention in the field of drug delivery because of their advantages, such as capable of entrapping both hydrophilic and hydrophobic drugs in their aqueous inner core and lipid bilayer, high stability, biocompatible, biodegradable [48,49]. Niosome is well-designed bilayer membrane carrier, but it takes several hours to release the drug [50]. On the other hand, the release of toxins from PLGA NPs is considerably faster and is expected to have a rapid drug effect under acidic environment." in Page 8, Lines 325-332, please kindly see it.

Comment: authors used some abbreviations without any explanation, it must be added

Response: Thank you for your kind comment. We added that explanations of abbreviations, please kindly see them.

Comment: why does DLS have a bigger size than TEM? Can u explain that?

Response: Thank you for your valuable comment. Particle size has a distribution, so there is an error between TEM and DLS. In DLS measurement, the nanoparticles were dispersed in aqueous solution. Meanwhile, in TEM observation, the nanoparticles were observed in dry condition. Therefore, nanoparticles may be shrunk during the dry process, resulting in a smaller size than that of DLS measurement.

Round 2

Reviewer 1 Report

The Authors replied to the raised points, and better explained some of the critical aspects, and acknowledged some of the limitations detected in the manuscript. Indeed they considered relevant the stability in a relevant biological environment (like plasma stability) and the use of co-cultured cells. The uncontrolled release of the toxins and stability of the final construct needs to be taken into account for a further evaluation and progression of the work. The limitation of the work being detected and acknowledged by the Authors needs to be briefly reported in the final conclusion and better conceived/argued in the discussion section.

Reviewer 2 Report

The authors answered some of my concerns about their manuscript. By the way, I don't agree with the authors about the different sizes of DLS and TEM. Please clarify this with some similar papers and also add this new explanation to the manuscript. Also, please discuss suggested references as comment 4. these interesting papers can improve the background of your paper.
